# Analysis of Thermomechanical Stresses of a Photovoltaic Panel Using a Passive System of Cooling

Brayan L. Pérez Escobar [1], Germán Pérez Hernández [1], Arturo Ocampo Ramírez [2], Lizeth Rojas Blanco [1], Laura L. Díaz Flores [1], Inocente Vidal Asencio [1], José G. Hernández Perez [3] and Erik Ramírez Morales [1,*]

1   División Académica de Ingeniería y Arquitectura, Universidad Juárez Autónoma de Tabasco, Cunduacán 86040, Mexico; leonardo.perez@ujat.mx (B.L.P.E.); german.perez@ujat.mx (G.P.H.); lizeth.rojas@ujat.mx (L.R.B.); laura.diaz@ujat.mx (L.L.D.F.); inocente.vidal@ujat.mx (I.V.A.)
2   Unidad Académica Profesional Tianguistenco, Universidad Autonoma del Estado de México, Paraje el Tejocote s/n, San Pedro Tlaltizapan 52640, Mexico; ingaor@hotmail.com
3   Industrias no Contaminantes, Engineering Faculty, Universidad Autonoma de Yucatan, A.P. 150, Merida 97000, Mexico; joshdez5@gmail.com
*   Correspondence: erik.ramirez@ujat.mx

**Abstract:** In this paper, the gradient temperature and the thermomechanical stresses of a photovoltaic panel has been studied with and without heatsink. For this purpose, a three-dimensional analysis was carried out. Accordingly, a heat transfer analysis was developed. The numerical results show a cooling close to 26.7% with the proposed triangle fins compared with the rectangular fins studied before by another author, and the temperature distribution was determined. With this information, the stress analysis was carried out in order to find the effect on the panel due to the thermomechanical stresses. The aluminium frame was restricted to move freely. The resulting stresses field established the magnitude of the alternative stresses, resulting in a 6.7% drop compared with a reference panel. The guidelines of IEC 61215 have to be take into account. Due to the results obtained, the use of this kind of system in desert conditions is desirable because of its high operational temperature and due to the increase in heat transfer by the fins.

**Keywords:** photovoltaic; thermomechanical; stresses

## 1. Introduction

One of the principal problems that the world is dealing with is the need for clean energy, thus, satisfying the energy demand around the world, and diminishing green house gases. Fabio et al. [1] mentioned that by the year 2030 there will be an increase in energy demand of up to 33%. Due to this, a photovoltaic system is a powerful key for resolving this problem. The performance of the Photovoltaic System (PVS) is based on solar irradiation, environment temperature, and so on. Taking into account the case of Mexico, the irradiance is considered to be from 5.6 to 6.1 kW/m$^2$ per day and stable, but the environmental temperature is variable and cannot be considered stable [2].

The temperature oscillates in a range between 10 °C and 26 °C in zones where average humidity reaches up to 95% and overall rainfall is from 100 to 4000 mm. This phenomenon causes different performance in the PVS across Mexico. Consequently, the energy conversion in the PVS in different climates and different parts of the world is about 70 to 80% of its nominal potential [3,4]. Mexico is experimenting with a huge increase in their use of PVSs, but the few systems installed generate about 60 to 70% of their design potential [5,6], under the international average. In a study carried out by Tushar M. and A. Dhoble [7], it was mentioned that the supply of electric energy is one of the most important requirements to take into account right away. The increase in the operational temperature of a solar cell causes a significant reduction in its efficiency of 0.4–0.5% per degree Celsius [1,8].

It is important to keep the lowest possible operating temperature in order to increase the output power and decrease the gradient temperature in its back panel [4]. The production of electricity by this type of method depends on determined physical properties such as solar radiation, heat losses, material technology and the photovoltaic panel's operational temperature [9], and only about 20% of irradiance received is converted to electricity [10]. Extensive research efforts have been carried out for this purpose. J.G. Hernandez et al. [11] carried out an investigation with passive heat sink, which could decrease the operational temperature by up to 7 °C. A method proposed to do this is by putting fins on the back panel; their main function is to reduce the temperature gradient as much as possible [12–14]. It is important to keep in mind that a fracture could occur on the panel back due to a rise in thermal stresses within the module because of the temperature [15,16]. The temperature could also induce micro cracks [17] and, thus, the integral structure has to be studied to assure its safe operation, taking into account the guidelines IEC 6215 [18,19]. Figure 1 depicts the methodology used to decrease the operational temperature. First of all, a conventional panel was chosen in order to obtain the thermal distribution, then a comparison of the geometry of rectangular and triangular fins was evaluated, taking into account the boundary conditions. Thus, a geometry was selected to find out the best fin configuration for both cases, through thermal and structural results. The methodology described in Figure 1 was used to design the geometry, evaluate and validate the mechanical behavior and, finally, obtain the optimal design. Figure 2 depicts the mesh independence. When elements reach an average of 25,000 elements, they begin to stabilize and are optimal for both cases, in their thermal and structural analysis.

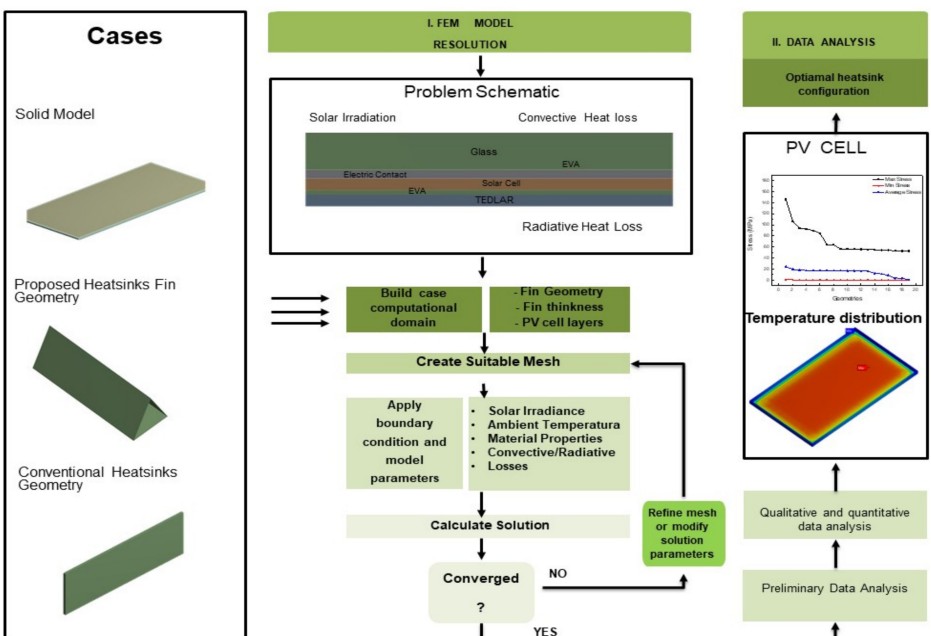

**Figure 1.** Flow chart which depicts the methodology used for decreasing the operational temperature in a solar panel.

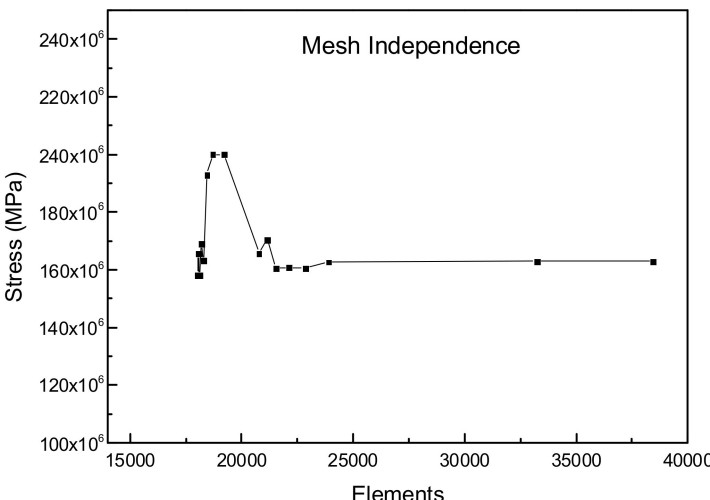

**Figure 2.** Mesh Independence.

The aim of this study was to register the stress of the electric contact subject to thermal loads and compare the traditional and the proposed heatsink, which increase the contact area of heat transfer and thus reduce the operational temperature. The use of aluminium fins can have a considerable impact on the lifetime and efficiency of the PV panel. This allowed us to monitor the behaviour of the fins and find the optimal design by evaluating different types of fin geometry, including their length and width. A numerical model in Ansys Workbench was used to study the effect of the design parameters.

## 2. Materials and Methods

Initially, the maximum temperature during a common day was determined using a conventional photovoltaic panel, then, modified panels with different thicknesses of EVA (from $2 \times 10^{-4}$ to $6 \times 10^{-4}$ m) were made, and finally aluminium fins were added as a passive cooling system (triangle fins with a thickness of 1 mm). For this purpose, a steady state thermal analysis was performed in order to find the distribution of temperature in the whole panel. A solid model of such a panel is shown in Figure 3. The field of temperature was used to determine the principal stresses through a structural analysis.

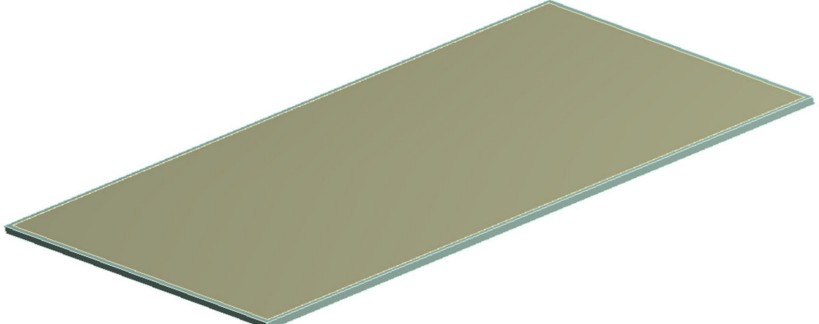

**Figure 3.** Solid model of the photovoltaic panel.

Thermomechanical Stress Analysis of the Photovoltaic Panel

The mesh used for the panel analysis has 38,446 elements and 284,198 nodes (see Figure 4). Figure 5 schematically shows the boundary conditions for the thermal analysis. Firstly, the room temperature began at 28 °C and the temperature was increased by 10 °C until it reached 40 °C. The top surface was subjected to a direct solar radiation of 1000 W/m$^2$, similar to the analysis carried out by [7,20,21] when the panel was exposed to a theorical film coefficient of 20 W/m$^2$ °C.

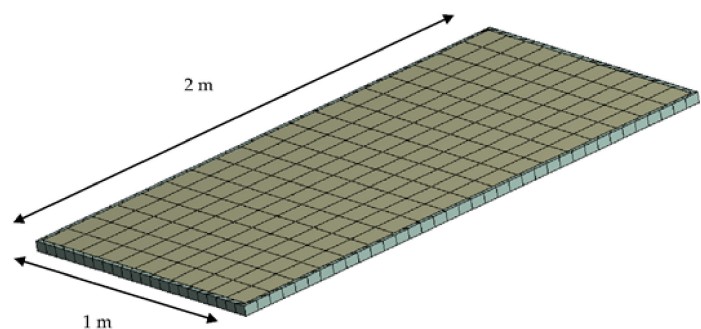

**Figure 4.** Mesh of the photovoltaic panel.

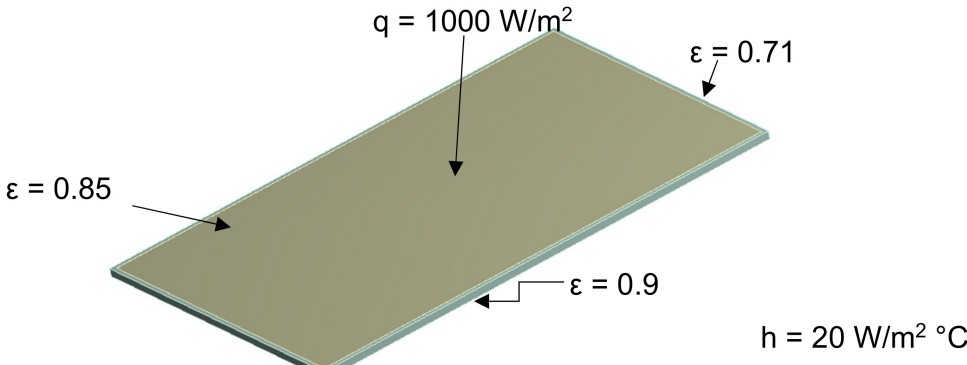

**Figure 5.** Boundary conditions used in the thermal analysis.

In the first step, a steady state thermal analysis was carried out. Its purpose was to simulate the operating temperature condition before the fins were added and with different thicknesses of EVA in order to improve the heat transfer as performed in the study carried out by El-Moneim et al. [22]. After this, different types of designs of fins were tested. Thus, after each design, the temperature back surface was obtained. Figure 6 represent the fin geometry studied.

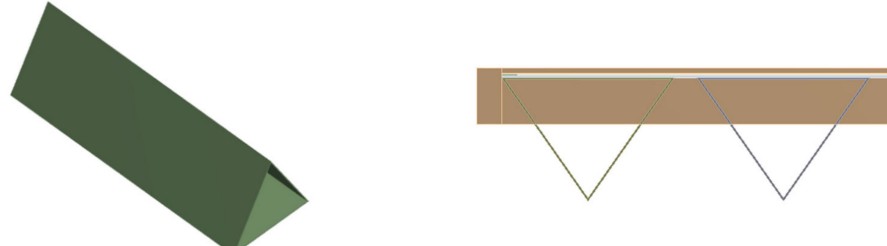

**Figure 6.** Geometry studied in the cases.

In all cases, the element selected for the FEM analysis was Solid 90. It can be applied to a three-dimensional steady state or transient thermal analysis. The material of the frame panel was aluminium, silicon was used for the solar cell, photovoltaic glass for top surface, EVA (ethylene vinyl acetate) as encapsulant and Tedlar as back sheet. The Table 1 shows the values of the properties of such materials.

**Table 1.** Material properties.

| Material | Thickness (mm) | Thermal Conductivity (W/mK) | Density kg/m³ | Coefficient of Thermal Expansion 1/°C | Specific Heat J/kg °C | Module of Young GPa | Ratio of Poisson | Tensile Yield Strength GPa |
|---|---|---|---|---|---|---|---|---|
| Glass | 3 | 1.8 | 2500 | $9 \times 10^{-6}$ | 795 | 0.7 | 0.2 | - |
| Silicon | 0.35 | 148 | 2330 | - | 677 | 168 | 0.3 | - |
| EVA | 0.5 | 0.35 | 960 | $0.73 \times 10^{-6}$ | 2090 | 2.1 | 0.4 | 0.00435 |
| Tedlar | 0.1 | 0.2 | 2700 | $0.73 \times 10^{-6}$ | 1000 | 2.1 | 0.4 | 0.11 |
| Aluminium | 35 | 209.3 | 2698.4 | $0.24 \times 10^{-6}$ | 900 | 70 | 0.33 | 2330 |
| Air | | 0.025 | - | | - | | | |

## 3. Results

The field of temperature and the maximum principal stress field when the peak heat flux is acting on the panel surface with a maximum environment temperature is shown in Figure 7. With respect to Figure 7a vs. Figure 7b, it can be seen that the temperature field was almost uniform on the whole panel body, except on the identified zone; a considerable decline in temperature operating of almost 3 °C resulted in decreasing the thickness of the EVA by 0.2 mm. It is important to keep in mind that this is such a thin film that it could lead to direct contact with the other materials (silicon and glass/Tedlar layers) but with advance preparation to enhance the properties of the materials this problem could be resolved [23].

Figure 7c shows schematically the field temperature distribution using rectangular fins (0.30 cm × 0.1 cm × 0.005 cm), having a maximum and a minimum temperature of 44.095 °C and 33.27 °C, respectively. Comparing Figure 7c with Figure 7d, it can been seen that there is a considerable drop in the maximum temperature and the back side keeps the temperature constant, while in Figure 7c a slight increase of 2 °C is shown.

In Figure 7d, we can appreciate the response of the aluminium triangle fins (5 cm × 5 cm × 30 cm, 1 mm of width) as a heatsink when a heat flux is acting directly on the whole front of the panel, as it shows a decrease of 2.316 °C in the operational temperature compared with Figure 7c. The temperature gradient on the back panel shows a slight improvement and a major uniform gradient due to it having more area for heat transfer. Thus, the panel obtains a better power output.

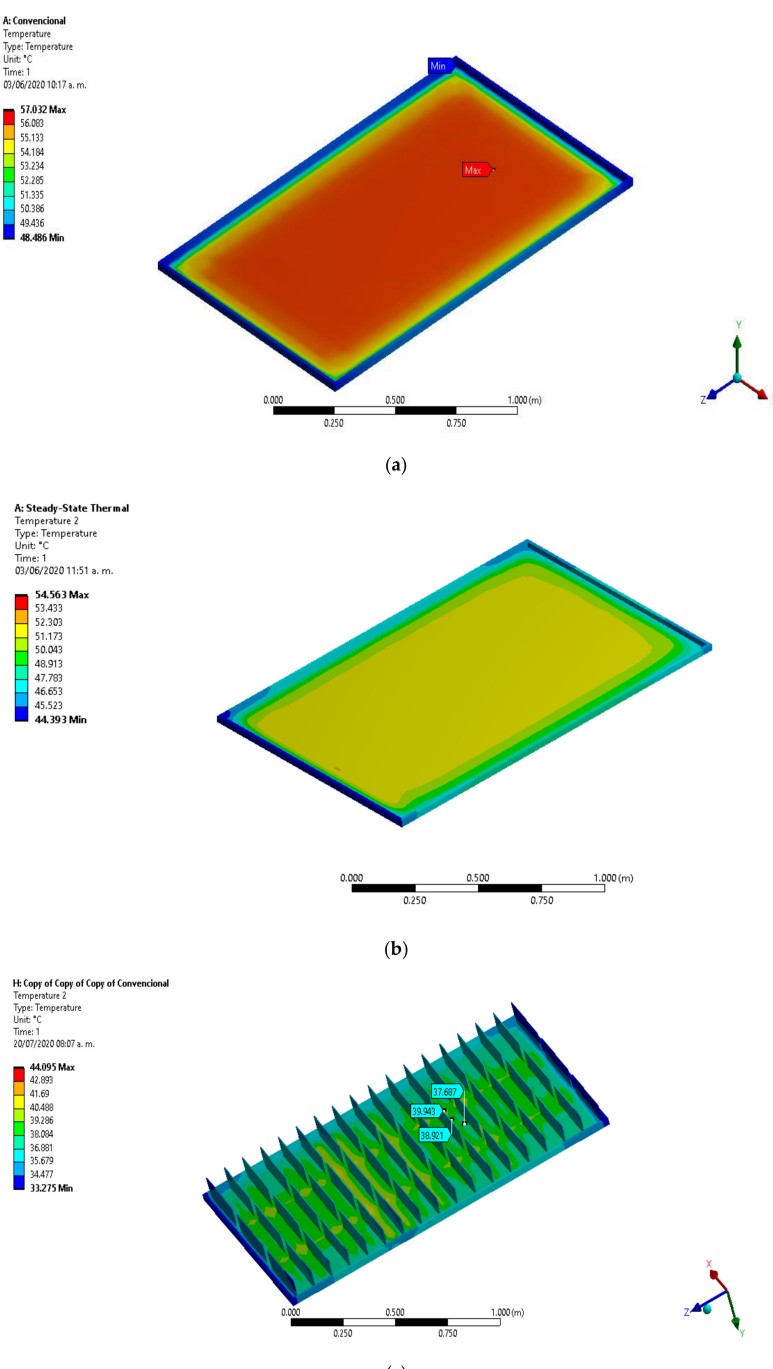

(**a**)

(**b**)

(**c**)

**Figure 7.** *Cont.*

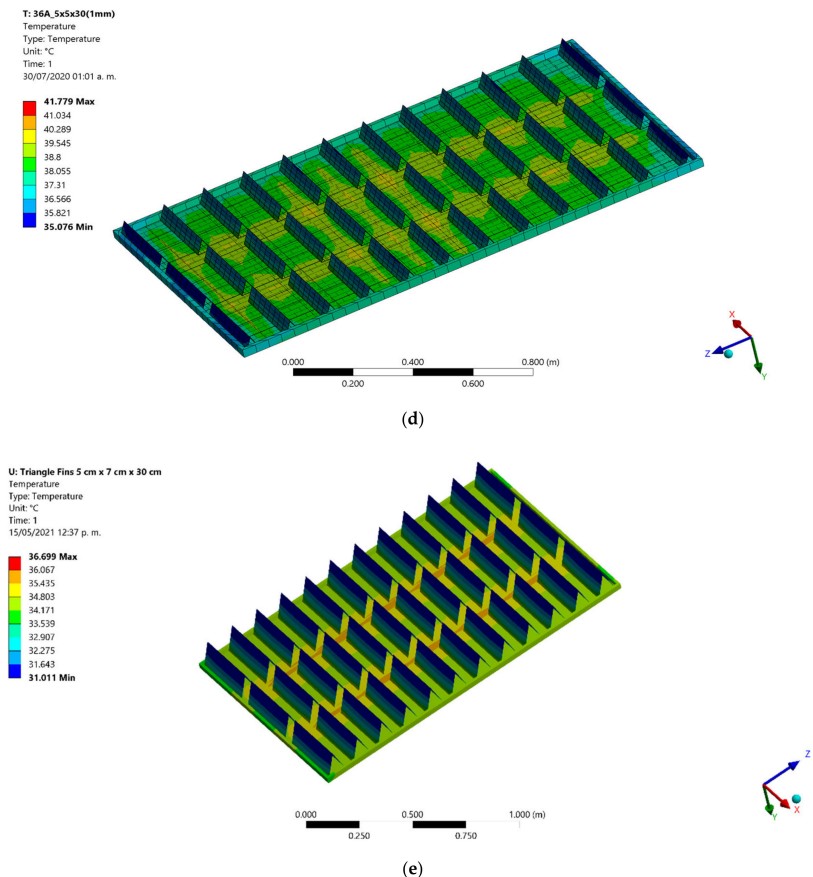

(d)

(e)

**Figure 7.** Temperature fields. (**a**) Temperature distribution of the whole panel EVA 0.5 mm. (**b**) Temperature distribution of the whole panel, EVA 0.2 mm. (**c**) Temperature distribution of the whole panel with rectangular fins. (**d**) Temperature distribution of the whole panel, EVA 0.2 mm. (**e**) Temperature distribution of the whole panel, fins 1 mm of thickness EVA 0.2 mm.

Regarding Figure 7e, a representative decline of 5.08 °C in the maximum operating temperature can be seen using fins with the following dimensions: 5 cm base × 7 cm high, 30 cm deep and a fin thickness of 1 mm. Meanwhile, its back temperature remains almost stable at 34 °C. This could be a good option for the fin's design; however, while the fin is higher, the temperature tends to go into a thermal equilibrium with the environment and stops cooling the panel. For that reason, this type of design is not a recommended option since the gradient rises.

In order to perform the stress analysis, the following considerations were taken into account: the aluminium frame was restricted to move freely and the earth's gravity was considered at the center of the panel.

Figure 8a shows the distribution of the equivalent stress of von Mises. The module experiences tensile stresses with the interconnection at the edges experiencing the highest tensile stress of 119.57 MPa. The mechanical stress at the corner is low because of the possibility of free expanding. Figure 8b shows the critical area, which represents the electric contact that, under high cycles, could lead to a fracture.

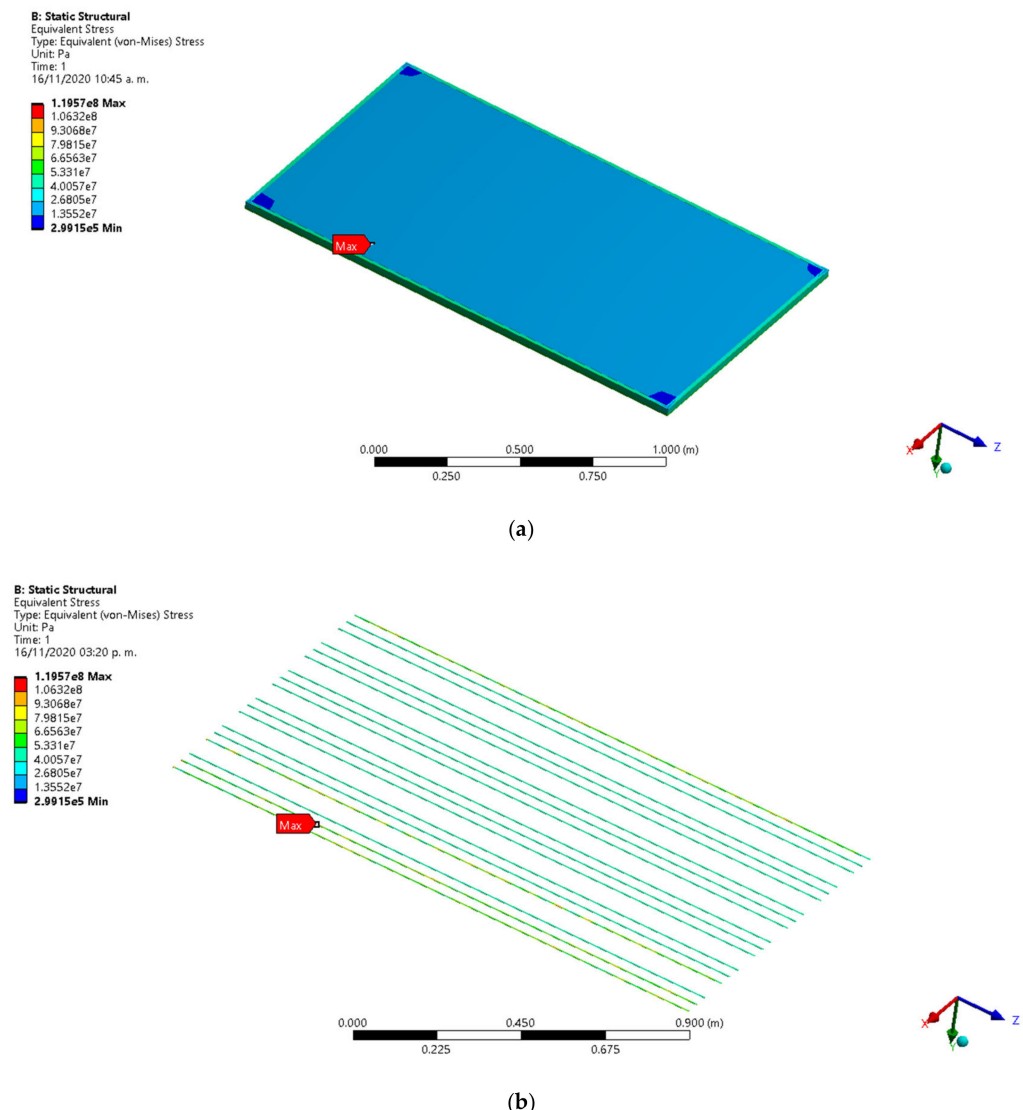

(**a**)

(**b**)

**Figure 8.** *Cont.*

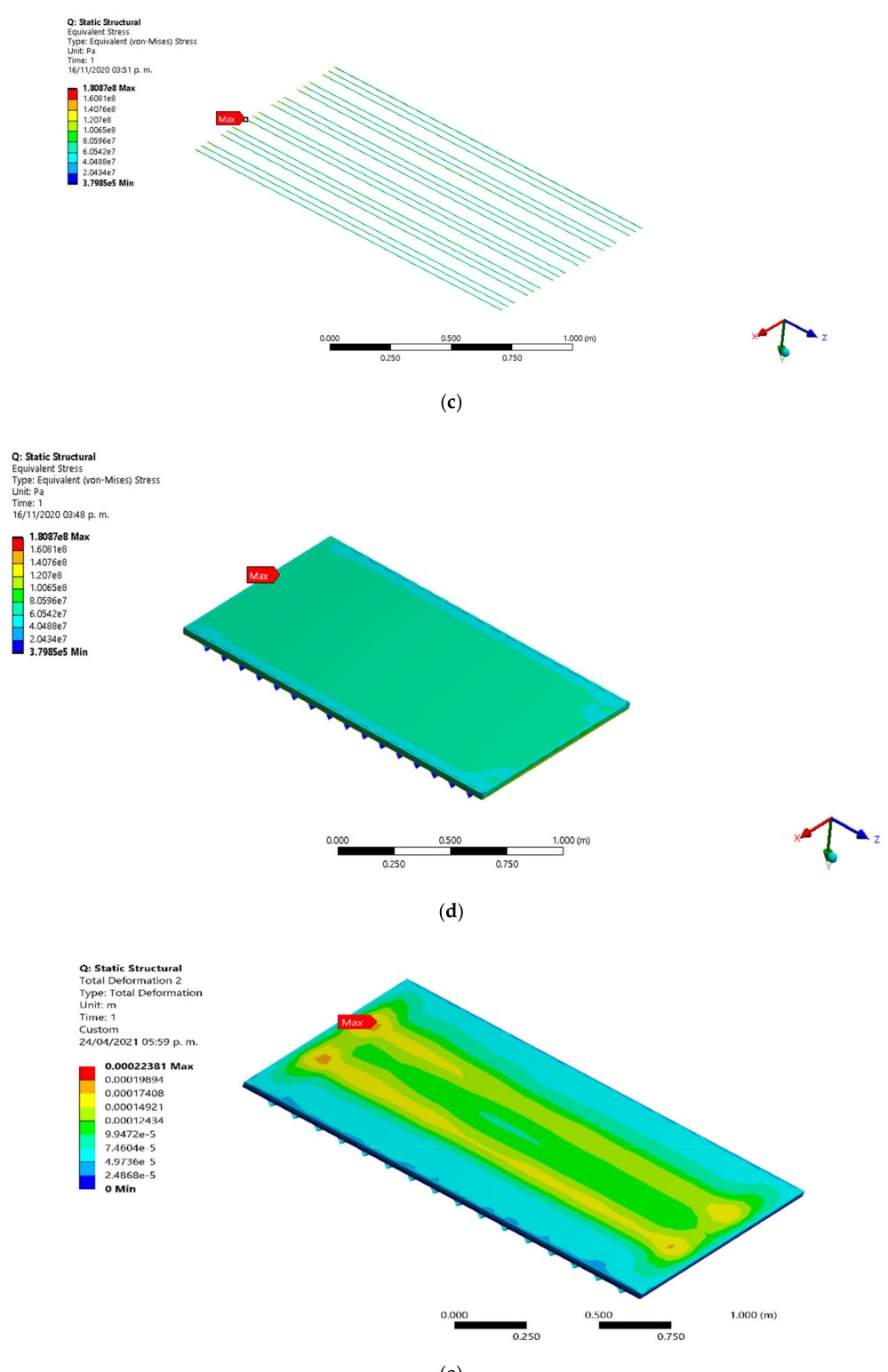

**Figure 8.** Maximum and minimum stress fields. (**a**) Stress distribution of the whole panel. (**b**) Stress distribution at the electric contact. (**c**) Stress distribution of the whole panel with fins. (**d**) Stress distribution at the electric contact with fins. (**e**) Total deformation at the panel.

Comparing Figure 8a with Figure 8c, a considerable increase in equivalent stress can be seen. This is because the fins on the back panel restrict the free movement. A 60 MPa increment is present in Figure 8b versus Figure 8d, which could lead to a fracture. In regard to Figure 8e, it can be seen there is 0.223 mm of displacement that generates the up stresses.

Figure 9 shows the results obtained on the back of the panel with the different models studied and Table 2 shows the parameters used.

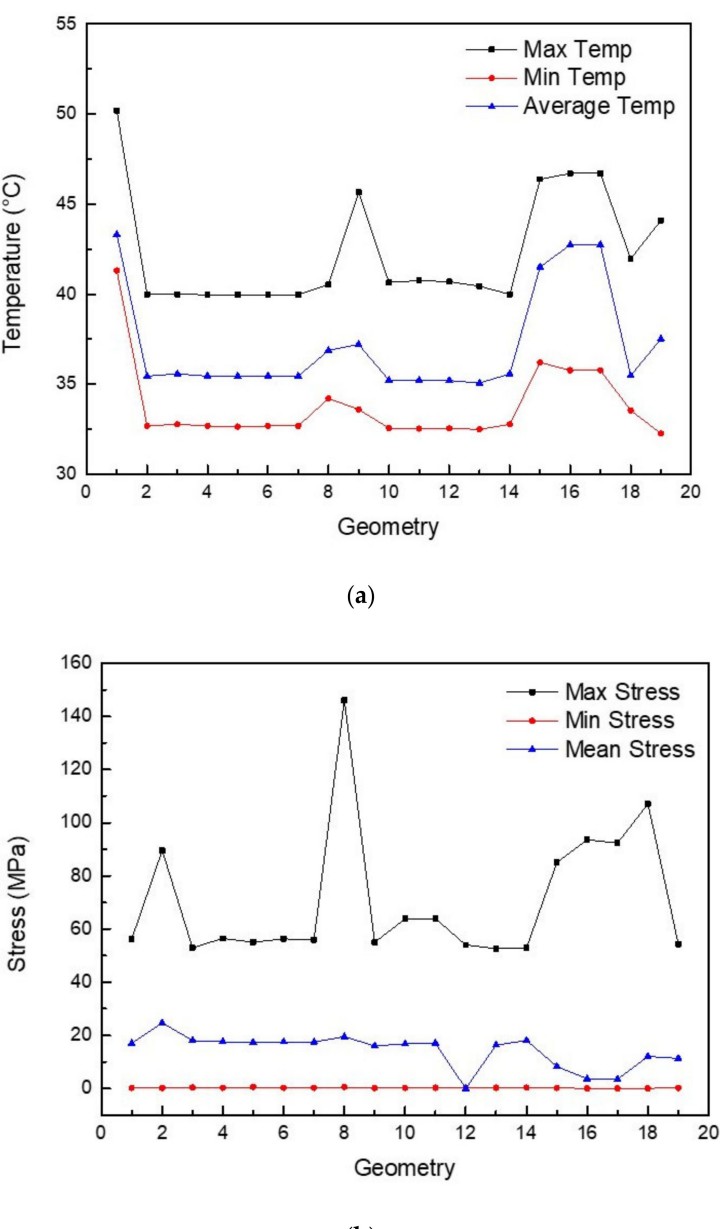

(**a**)

(**b**)

**Figure 9.** Results of temperature and stresses of the studied cases. (**a**) Field temperature distribution of the geometries studied; (**b**) Field stress distribution of the geometries studied.

**Table 2.** Results of different designs.

| Parameters | Values |
|---|---|
| Convection | 20 W/m$^2$ °C @ 30 °C |
| Radiation | 0.85 @ 30 °C |
| Radiation 2 | 0.90 @ 30 °C |
| Radiation 3 | 0.71 @ 30 °C |
| Heat flux | 1000 W/m$^2$ |

In Figure 9, it can been seen that there were 19 geometries studied with different configurations (Table 3). Initially, the conventional panel had a maximum temperature and stresses of 50.19 °C and 56.3 MPa, respectively. Secondly, an array with 51 fins of 5 cm × 5 cm × 1 m and 1 mm of width was tested; a considerable drop was seen of almost 10 °C and stresses remained stable. Then, 24 triangle fins maintained a mean temperature, but an important increase in stress was shown. From 10 to 14, an array with 48 triangle fins presented a temperature average of 40 °C and a maximum stress of 63.87 MPa. Finally, a stress of 107.11 MPa was presented with 53 triangle fins of 5 cm × 5 cm × 30 cm and 1 mm of width.

**Table 3.** Geometries Studied.

| Geometry | Fins | Fin Number |
|---|---|---|
| 1 | Convectional panel | Conventional Panel |
| 2 | Rectangular | 51 fins of 30 cm × 10 cm × 0.005 cm |
| 3 | Rectangular | 14 fins of 1 m × 10 cm × 0.005 cm EVA $5 \times 10^{-4}$ m, TEDLAR $4 \times 10^{-4}$ m |
| 4 | Triangle | 14 fins of 5 cm × 5 cm × 1 m, EVA $5 \times 10^{-4}$ m, $5 \times 10^{-4}$ m, TEDLAR $4 \times 10^{-4}$ m |
| 5 | Triangle | 14 fins of 5 cm × 5 cm × 1 m, TEDLAR $2 \times 10^{-4}$m, EVA $2 \times 10^{-4}$m, $2 \times 10^{-4}$ m |
| 6 | Triangle | 14 fins of 5cm × 5 cm × 1 m, EVA $4 \times 10^{-4}$, TEDLAR $5 \times 10^{-4}$ m |
| 7 | Triangle | 14 fins of 5 cm × 5 cm × 1 m, EVA $3 \times 10^{-4}$ m, $3 \times 10^{-4}$ m, TEDLAR $5 \times 10^{-4}$ m |
| 8 | Triangle | 24 fins of 1 m × 10 cm × 5 cm, TEDLAR $4 \times 10^{-4}$ m, EVA $5 \times 10^{-4}$ m, $5 \times 10^{-4}$ m |
| 9 | Triangle | 36 fins of 5 cm × 5 cm × 30 cm, EVA $2 \times 10^{-4}$ m, TEDLAR $5 \times 10^{-4}$ |
| 10 | Triangle | 48 fins of 5 cm × 5 cm × 30 cm |
| 11 | Triangle | 48 fins of 5 cm × 5 cm × 30 cm, TEDLAR $5 \times 10^{-4}$ m, EVA $6 \times 10^{-4}$ m, $6 \times 10^{-4}$ m |
| 12 | Triangle | 48 fins of 5 cm × 5 cm × 30 cm, TEDLAR $5 \times 10^{-4}$ m, EVA $4 \times 10^{-4}$ m, $4 \times 10^{-4}$ m |
| 13 | Triangle | 48 fins of 5 cm × 5 cm × 30 cm, EVA $3 \times 10^{-4}$ m, $2 \times 10^{-4}$ m, TEDLAR $5 \times 10^{-4}$ m |
| 14 | Triangle | 48 fins of 5 cm × 5 cm × 30 cm, TEDLAR $5 \times 10^{-4}$ m, EVA $6 \times 10^{-4}$ m, $6 \times 10^{-4}$ m |
| 15 | Triangle | 51 fins of 5 cm × 5 cm × 30 cm, EVA, TEDLAR $5 \times 10^{-4}$ m |
| 16 | Triangle | 51 fins of 5 cm × 5 cm × 30 cm, TEDLAR $5 \times 10^{-4}$ m, EVA $2 \times 10^{-4}$ m |
| 17 | Triangle | 51 fins of 5 cm × 5 cm × 30 cm, EVA TEDLAR $5 \times 10^{-4}$ m |
| 18 | Triangle | 53 fins of 5 cm × 5 cm × 30 cm, EVA de $2 \times 10^{-4}$ m, TEDLAR $5 \times 10^{-4}$ m, 28 holes |
| 19 | Triangle | 54 fins of 5 cm × 5 cm × 30 cm, EVA $2 \times 10^{-4}$ m, TEDLAR $5 \times 10^{-4}$ m |

## 4. Conclusions

As can been seen, the aim of this study was to reduce the operational temperature in order to increase the output power and, thus, increase the life of a panel. To reach this aim, the use of fins on the back panel as a heatsink was studied. Due to this, the free movement of the materials were restricted and thermomechanical stresses appeared.

The maximum stress when there are 36 fins as a heatsink is 146.11 MPa, the most critical of the analysed cases. It is noteworthy that although the use of fins as a heatsink in a photovoltaic panel can reduce significantly the operational temperature, micro cracks could

be induced because of high thermomechanical stresses and this would cause a considerable drop in the output power.

To assure the effective operation of the photovoltaic panel, the recommendations mentioned in the guidelines of IEC 61215 should be taken into account, some of which are regarding monitoring the temperature coefficient and the thermal cycle.

The manufacture of heat sinks should be performed in order to carry out an experimental analysis and, thus, compare the numerical and experimental results.

**Author Contributions:** Conceptualization, B.L.P.E. and E.R.M.; methodology, L.L.D.F. and I.V.A.; software, A.O.R.; validation, B.L.P.E. and J.G.H.P.; formal analysis, L.R.B.; investigation, E.R.M.; writing—original draft preparation, B.L.P.E.; writing—review and editing, L.L.D.F., A.O.R. and G.P.H. visualization, G.P.H. All authors have read and agreed to the published version of the manuscript.

**Funding:** This research received no external funding.

**Institutional Review Board Statement:** Not applicable.

**Informed Consent Statement:** Not applicable.

**Data Availability Statement:** The data used to support the findings of this study are available from the corresponding author upon request.

**Conflicts of Interest:** The authors declare no conflict of interest.

## Symbology

Symbology used at the document

| IEC | International Electrotechnical Commission |
|-----|-------------------------------------------|
| PVS | Photovoltaic System |
| PV | Photovoltaic |
| EVA | Ethylene Vinyl Acetate |
| FEM | Finite Element Analysis |

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
