# Peer review of "Analysis of Thermomechanical Stresses of a Photovoltaic Panel Using a Passive System of Cooling"

_applsci, doi:10.3390/app11219806_

Round 1

Reviewer 1 Report

Significant scientific contribution to possible practical application. It is desirable in the future work to develop for other forms of photovoltaic panels even greater efficiency.

Reviewer 2 Report

By changing the heat sink structure, increasing the contact area, reducing the operating temperature, thereby increasing the output power of the photovoltaic system, and simulates the surface thermal stress distribution, ensuring that the entire system can safely and stable, The research has clear thinking, complete content and strong engineering application. However, some problems should be addressed.

  1. The most important factor affecting solar power generation is irradiance. Especially in different seasons, the irradiance fluctuates significantly in one day, and the change of irradiance also has a great influence on the ambient temperature. Please give convincing reasons: prove that irradiance does not have too much influence on the research of thepaper.

  1. There are various types of radiators, and it is hoped that a comparative analysis can be made with other types of radiators, so as to highlight the advantages of the structure proposed by the author.

  1. Please explain in detail the physical meaning of the symbols in the passage.

  1. Please explain that in the boundary condition of Figure 6, ε = 0.85, ε = 0.71 represents which surface in the model?

Reviewer 3 Report

I suggest the following:
- restoring table 3 to make it easier to read;
- checking the units of measurement (eg MPa)
- in Conclusion please provide some information about future trends and whether it is possible to scale the process to an industrial scale so that it can be applied in practice.

After those minor changes, the paper can be publishable.
